# On Joint Noise Scaling in Differentially Private Federated Learning with Multiple Local Steps

## Abstract

Federated learning is a distributed learning setting where the main aim is to train machine learning models without having to share raw data but only what is required for learning. To guarantee training data privacy and high-utility models, differential privacy and secure aggregation techniques are often combined with federated learning. However, with fine-grained protection granularities the currently existing techniques require the parties to communicate for each local optimization step, if they want to fully benefit from the secure aggregation in terms of the resulting formal privacy guarantees. In this paper, we show how a simple new analysis allows the parties to perform multiple local optimization steps while still benefiting from joint noise scaling when using secure aggregation. We show that our analysis enables higher utility models with guaranteed privacy protection under limited number of communication rounds.

## 1 Introduction

Federated learning (FL; McMahan et al. 2017; Kairouz et al. 2019) is a common distributed learning setting, where a central server and several clients holding their own local data sets collaborate to train a single global model. The main feature in FL is that the clients do not directly communicate data, but only what is required for learning, e.g., gradients or updated model parameters (pseudo-gradients).

While FL satisfies the data minimization principle, i.e., only what is actually needed is communicated while the actual raw data never leaves the client, it does not protect against privacy attacks such as membership inference (Shokri et al., 2017) or reconstruction (Fredrikson et al., 2014; Yeom et al., 2018). Instead, training data privacy is commonly ensured by combining differential privacy (DP; Dwork et al. 2006b), a formal privacy definition, and secure multiparty computation (MPC; Yao 1982) with FL (see, e.g., Kairouz et al. 2019).

DP is essentially a robustness guarantee for stochastic algorithms, which guarantees that small perturbations to the inputs have small effects on the algorithms' output probabilities. What constitutes a small perturbation depends on the chosen protection granularity: the same basic DP definition can be used for ensuring privacy on anything from single sample to entire data set level. In turn, MPC protocols can be used to limit the amount of information an adversary has about computations. In FL, secure aggregation (SecAgg) protocols, a specialised form of secure computation that requires significantly less resources than general MPC, are commonly used for communicating model updates from the clients to the server, which can result in provably better joint DP guarantees than is possible to achieve by any single client in isolation.

Under the general FL setup, two main alternatives are commonly considered: cross-device FL and cross-silo FL (Kairouz et al., 2019). In cross-device FL, each client is assumed to have a small local data set, while the total number of clients is large, e.g., thousands or millions. In the cross-silo case, the total number of clients is small, for example, a dozen, but each client is assumed to have a larger local data set. In this paper, our running example is standard cross-silo differentially private FL (DPFL) where the clients communicate all updates to the server using SecAgg.[1] In this setting,

---

[1]Instead of considering any specific SecAgg implementation, in this work we mostly assume an idealised trusted aggregator. We discuss practical implementations in Appendix A.2.

the most useful DP protection granularity is typically something strictly more fine-grained than client-level: when clients are, e.g., different hospitals or banks, there are typically several individuals in a single clients' local data set and the protection granularity needs to match the use case.

While client-level granularity in DPFL is, at least in principle, straightforward to combine with SecAgg, more fine-grained granularities such as sample-level DP can present problems: using existing techniques one has to choose between i) having joint DP guarantees with less noise due to SecAgg but with all clients using only a single local optimization step per FL round, and ii) having more noisy local DP (LDP) guarantees that do not formally benefit from SecAgg while allowing the clients to do more local optimization steps per FL round. Both of these options have significant drawbacks: the amount of server-client communications is typically one of the first bottlenecks that limit model training in FL, while LDP guarantees regularly require noise levels that heavily affect the resulting model utility. In this paper we show that this trade-off is not unavoidable but can be largely remedied by a simple new analysis of the problem.

**Our Contribution**

- We present a novel and simple theoretical privacy analysis showing when we can increase the number of local optimization steps in FL using fine-grained DP granularity, while still benefiting from joint DP guarantees using a trusted aggregator.

- We demonstrate empirically that the proposed approach can lead to large utility benefits without requiring any changes to the underlying algorithms under both iid and heterogeneous client data splits.

- Our results point to a clear mismatch between the current theoretical understanding of DPFL and practical results.

## 2 RELATED WORK

There is a significant amount of existing work focusing on the general problem of combining DP with FL, although the focus has mostly been on the cross-device FL setting with user- or client-level DP. To the best of our knowledge, while the combination of DPFL with SecAgg is certainly not novel (see, e.g., Truex et al. 2019; Kairouz et al. 2019; Heikkilä et al. 2020; Stevens et al. 2022; Yang et al. 2023), there is no existing work on the privacy analysis when the clients do multiple local optimization steps with fine-grained DP and communicate the results via SecAgg.[2]

Considering the existing work in more detail, we can distinguish some main lines of closely-related research. There are many papers proposing novel learning methods for FL, assuming sample-level DP and joint noise scaling with SecAgg. While the existing work only uses a single local optimization step (see, e.g., Heikkilä et al. 2020; Malekzadeh et al. 2021; Stevens et al. 2022; Yang et al. 2023), our analysis can be leveraged in this setting to enable running multiple local steps generally for many such methods without requiring any other changes to the algorithms.

Another clear line of work has focused on introducing novel discrete DP mechanisms that can be used with additively homomorphic encryption techniques, which typically operate on the group of integers with modulo additions. Agarwal et al. (2018) proposed a binomial mechanism that provides DP using discrete binomial noise. Improving on the binomial mechanism, Canonne et al. (2020) proposed a discrete Gaussian mechanism, while Agarwal et al. (2021) introduced a Skellam mechanism and Chen et al. (2022b) a Poisson-binomial mechanism, both of which improve on the discrete Gaussian, e.g., by being infinitely divisible distributions: the sum of Skellam/Poisson-binomial distributed random variables is another Skellam/Poisson-binomial random variable. Our work is not focused on introducing new DP mechanisms, but our analysis allows for using many different DP noise mechanism. In particular, our analysis allows for joint noise scaling under SecAgg including when using infinitely divisible DP mechanisms, such as the Skellam mechanism, with pseudo-gradients and fine-grained DP protection level.

---

[2]Note that (Truex et al., 2019, Algorithm 4) seems to state a weaker, specialised version of our results, i.e., they use several local optimization steps with sample-level DP and SecAgg in FL, while scaling the noise jointly over the clients. However, as also noted by Malekzadeh et al. (2021), the approach of Truex et al. (2019) would require a separate proof of privacy beyond what is actually provided in the paper.

While our main focus is on privacy accounting with SecAgg under limited communication budget, there has also been considerable effort by the community to reduce the amount of required communication further by applying quantization to the gradients (Agarwal et al., 2018; Kairouz et al., 2021; Agarwal et al., 2021; Chen et al., 2022b; Jin et al., 2020; Chaudhuri et al., 2022; Guo et al., 2023) or by compressing the updates sent by the clients (Triastcyn et al., 2021; Chen et al., 2022a). In principle, any such technique for compressing the model updates compatible with SecAgg can also be directly combined with our joint noise scaling analysis. In contrast, benefiting from gradient quantization is not entirely straightforward as in our case the model updates are pseudo-gradients and not gradients. We leave a detailed consideration and comparison of the possible methods for reducing the required communication budget beyond what is possible by pushing more optimization steps to the clients for future work.

In summary, while many of the contributions cited above, e.g., novel DP mechanisms, are not limited to cross-device FL, all the experiments and use cases mentioned in the cited papers that are compatible with SecAgg and use multiple local steps only consider joint noise scaling with *user- or client-level DP* in cross-device FL. In contrast, we focus on more fine-grained DP granularities, namely on *sample-level DP*. As we discuss in Section 3, combining sample-level DP with multiple local steps and joint noise scaling using SecAgg with good utility requires a novel privacy analysis. The main aim of this paper is to provide such an analysis.

While the currently existing theoretical convergence bounds for DPFL do not show any benefit from increasing the number of local steps in DPFL (see Malekmohammadi et al. 2024, Theorem 3.2), we empirically demonstrate the utility of our analysis in Section 5 after stating the results in Section 4. Our results clearly highlight the need for improving the theoretical analysis of DPFL over what is shown by Malekmohammadi et al. (2024) to understand when increasing the number of local steps is useful (compare this disagreement of empirical results and theory to the discussion by Mishchenko et al. 2022 on the provable usefulness of local steps in non-DP FL).

## 3 BACKGROUND

Federated learning (FL, McMahan et al. 2017; Kairouz et al. 2019) is a collaborative learning setting, where the participants include a central server and clients holding some data. On each FL round, the server chooses a group of clients for an update and sends them the current model parameters. The chosen clients update their local model parameters by taking some amount of optimization steps using only their own local data, and then send an update back to the server. The server then aggregates the client-specific contributions to update the global model. We use the standard federated averaging update rule: assuming w.l.o.g. that clients $i = 1, \ldots, N$ have been selected at FL round $t$, and that client $i$ sends an update $\Delta_i^{(t)}$ (pseudo-gradient), the updated global model $\theta_t$ is given by

$$\theta_t = \theta_{t-1} + \frac{1}{N} \sum_{i=1}^{N} \Delta_i^{(t)}. \tag{1}$$

### 3.1 DIFFERENTIAL PRIVACY

We want to guarantee privacy of the trained model w.r.t. the training data, for which we use differential privacy (DP). Writing the space of possible data sets as $\mathcal{X}^* := \cup_{n \in \mathbb{N}} \mathcal{X}^n$, we have the following:

**Definition 3.1.** (Dwork et al., 2006b;a) Let $\varepsilon > 0$ and $\delta \in [0, 1]$. A randomised algorithm $\mathcal{A} : \mathcal{X}^* \to \mathcal{O}$ is $(\varepsilon, \delta)$-DP if for every $x, x' \in \mathcal{X}^* : x \simeq x'$, and every measurable set $E \subset \mathcal{O}$,

$$\mathbb{P}(\mathcal{A}(x) \in E) \leq e^\varepsilon \mathbb{P}(\mathcal{A}(x') \in E) + \delta,$$

where $\simeq$ is a neighbourhood relation. $\mathcal{A}$ is tightly $(\varepsilon, \delta)$-DP, if there does not exist $\delta' < \delta$ such that $\mathcal{A}$ is $(\varepsilon, \delta')$-DP. When $\delta = 0$, we write $\varepsilon$-DP and call it *pure DP*. The more general case $(\varepsilon, \delta)$-DP is called *approximate DP* (ADP).

Definition 3.1 can be equivalently stated as a bound on the so-called hockey-stick divergence:

**Definition 3.2.** Let $\alpha > 0$. The hockey-stick divergence between distributions $P, Q$ is given by

$$H_\alpha(P\|Q) := \mathbb{E}_{t \sim Q}\left(\left[\frac{dP}{dQ}(t) - \alpha\right]_+\right) = \mathbb{E}_{t \sim Q}\left(\left[\frac{p(t)}{q(t)} - \alpha\right]_+\right), \tag{2}$$

where $[a]_+ := \max(a, 0)$, $\frac{dP}{dQ}$ is the Radon-Nikodym derivative, and $p, q$ are the densities of $P, Q$, respectively. In the rest of this paper, we assume that all the relevant densities exists.

It has been shown that a randomised algorithm $\mathcal{A}$ is $(\varepsilon, \delta)$-DP iff $\sup_{x \simeq x'} H_{e^\varepsilon}(\mathcal{A}(x) \| \mathcal{A}(x')) \leq \delta$ (Barthe et al., 2013; Barthe & Olmedo, 2013).

Our main results do not depend on the exact neighbourhood definition, but in all the experiments we use the add/remove relation or unbounded DP, that is, $x, x' \in \mathcal{X}^*$ are neighbours, if $x$ can be transformed into $x'$ by adding or removing a single protected unit from $\mathcal{X}$. For the protection granularity, although this is not a strict limitation, we focus on the common sample-level DP, i.e., a single protected unit corresponds to a single sample of data. As noted earlier, our analysis could be advantageous for anything more fine-grained than client-level DP, e.g., element-level DP (Asi et al., 2019) or even for individual-level when there are more than one individual in the clients' local data.

We also make use of dominating pairs:

**Definition 3.3.** (Zhu et al., 2022) A pair of distributions $(P, Q)$ is a dominating pair for a stochastic algorithm $\mathcal{A}$, if for all $\alpha \geq 0$

$$\sup_{x, x' \in \mathcal{X}^*: x \simeq x'} H_\alpha(\mathcal{A}(x) \| \mathcal{A}(x')) \leq H_\alpha(P \| Q), \tag{3}$$

where $H_\alpha$ is the hockey-stick divergence (Definition 3.2).

## 3.2 Problem with Local Steps in DPFL with SecAgg

While DP offers strict privacy protection, it comes at the cost of reduced model utility. This is especially true in the local DP (LDP) setting, where each client protects its own data independently of any other party (Kasiviswanathan et al., 2008). One well-known technique to improve model utility in DPFL has been to utilise secure aggregation (SecAgg) to turn LDP guarantees into joint guarantees or distributed DP guarantees that depend on multiple clients (see, e.g., Kairouz et al. 2019). However, naively combining fine-grained DP protection with SecAgg for distributed DP runs into problems, as we demonstrate in the rest of this section.

Starting with the unproblematic case of client-level DP, writing $TA$ for an ideal trusted aggregator and using the well-known Gaussian mechanism (Dwork et al., 2006a) for simplicity, one can get joint DP guarantees for any number of local optimization steps with the following update:

$$\theta_t = \theta_{t-1} + \frac{1}{N} TA \left( \sum_{i=1}^N clip_C(\Delta_i^{(t)}) + \xi_i^{(t)} \right), \tag{4}$$

where the sum inside $TA$ is done by a trusted aggregator, $clip_C$ ensures that each client-specific update has bounded $\ell_2$-norm, and $\xi_i^{(t)}$ is Gaussian noise s.t. $\sum_{i=1}^N \xi_i^{(t)}$ gives the joint DP protection level we are aiming for. As the clipping and noise are applied directly to the updated weights after the local optimization has finished, the privacy protection is not affected by the number of local optimization steps client $i$ is using to arrive at $\Delta_i^{(t)}$ before applying DP.

There is also a simple approach that works with more fine-grained granularities, when the clients use a single local optimization step with common learning rate $\gamma$ and, for example, standard DP stochastic gradient descent (DP-SGD, Song et al. 2013) again utilising Gaussian noise: we can take $\Delta_i^{(t)} = -\gamma(g_i^{(t)} + \xi_i^{(t)})$, where $g_i^{(t)}$ is a sum of clipped per-unit gradients (e.g. per-sample for sample-level DP) from client $i$, to have the update

$$\theta_t = \theta_{t-1} - \frac{1}{N} TA \left( \sum_{i=1}^N \gamma(g_i^{(t)} + \xi_i^{(t)}) \right). \tag{5}$$

Looking at the sum in Equation 5, since each per-unit gradient has a common bounded norm and Gaussian noise is infinitely divisible, i.e., the summed-up noise is another Gaussian, we can calculate the resulting privacy with standard techniques (see, e.g., Mironov et al. 2019; Koskela et al. 2020; Zhu et al. 2022). Now, if one tries to use the same reasoning with sample-level DP using $S > 1$ local optimization steps, the problem is that the sensitivity of the per-sample clipped gradients when

summed over the local steps increases with $S$: assuming $\|g_{i,s}\|_2 \leq C, s = 1, \ldots, S$ implies that $\|\sum_{s=1}^{S} g_{i,s}\|_2 \leq SC$ (triangle-inequality).

In other words, trying to scale the noise over multiple local optimization steps naively ends up scaling the query sensitivity linearly with the total number of steps, while the obvious problem in using only a single step per FL round is that the number of communication rounds is typically one of the main bottlenecks in FL (Kairouz et al., 2019).

### 3.3 TRUST MODEL

In this paper, we assume an honest-but-curious (hbc) server and that all the clients are fully honest. The latter assumption can be easily generalised to allow for hbc clients with some weakening to the relevant privacy bounds: with $N$ (non-colluding) hbc clients, since any client could potentially remove its own noise from the aggregated results, the noise from the other $N - 1$ clients needs to guarantee the target DP level. In effect, to allow for all hbc clients, we would need to scale up the noise level somewhat (see, e.g., Heikkilä et al. 2017 for a discussion on noise scaling and for formal proofs).

In principle, the same technique can also protect against privacy threats in the case of including some fully malicious clients in the protocol (i.e, simply scale the noise so that the hbc clients are enough to guarantee the required DP level). However, in this case the required level of extra noise will increase quickly with the number of malicious clients leading to heavier utility loss. With malicious clients, there would also be no guarantee that the learning algorithm terminates properly.

## 4 JOINT NOISE CALIBRATION WITH MULTIPLE LOCAL STEPS USING A TRUSTED AGGREGATOR

Consider standard FL setting with $M$ clients and client $i$ holding some local data $x_i$. On FL round $t$, $N_t$ clients are selected for updating by the server, w.l.o.g. assumed to be clients $i = 1, \ldots, N_t$. Each selected client $i$ receives the current model parameters $\theta^{(t-1)}$ from the server, then runs $S_t$ local optimization steps using DP-SGD with constant learning rate $\gamma_t$, and finally sends an update to the server via a trusted aggregator $TA$:

$$\Delta_i^{(t)} = \theta_i^{(t)} - \theta^{(t-1)} = -\sum_{s=1}^{S_t} \gamma_t (g_{i,s}^{(t)} + \xi_{i,s}^{(t)}), \tag{6}$$

where we write $g_{i,s}^{(t)}$ for the per-unit clipped gradients of client $i$ at local step $s$, and $\xi_{i,s}^{(t)}$ for the DP noise. After receiving all the messages via the trusted aggregator, the server updates the global model using FedAvg:

$$\theta^{(t)} = \theta^{(t-1)} + \frac{1}{N_t} TA(\sum_{i=1}^{N_t} \Delta_i^{(t)}). \tag{7}$$

In the rest of this section we state our main results: we show that under some assumptions we can account for privacy in FL by looking at the local optimization steps while scaling the noise level jointly over the clients, even if there is no communication between the clients during the local optimization but only a single trusted aggregation at the end of the round to update the global model parameters.

W.l.o.g. from now on we drop the FL round index $t$ and simply write, e.g., $N$ instead of $N_t$ for the number of updating clients. Since the global updates do not access any sensitive data, once we can do privacy accounting for a single FL round, which is the main topic in the rest of this section, generalising to $T$ FL rounds can be done in a straightforward manner (see Appendix A.3).

In the following, we assume that all clients have access to an ideal trusted aggregator, and that all sums are calculated by calling the trusted aggregator. We comment on more realistic implementations in Appendix A.2 after stating our main results.

We make the following assumptions throughout this section:

**Assumption 4.1.** Let $x_i \in \mathcal{X}^*, i = 1, \ldots, N$. We write $x = \cup_{i=1}^N x_i$, and assume that $x_i \cap x_j = \varnothing$ for every $i \neq j$, i.e., there are no overlapping samples in different clients' local data sets. We are interested in fixed-length optimization runs of $S$ local steps (common to all clients), which leads to (fixed-length) adaptive sequential composition for privacy accounting (see e.g. Rogers et al. 2016; Zhu et al. 2022). We assume all clients use the same learning rate $\gamma$ and norm clipping with constant $C$ when applicable. We also assume that all local DP mechanisms $\mathcal{A}_i^{(s)}, s = 1, \ldots, S, i = 1, \ldots, N$ are DP w.r.t. the first argument for any given auxiliary values (which we generally do not write out explicitly).

Note that we consider how to loosen many of these assumptions in Appendix A.1.

Not all possible DP mechanisms might allow for joint noise scaling via simple aggregation. For convenience, in Definition 4.2 we define a family of suitable mechanisms, which we call sum-dominating:

**Definition 4.2** (Sum-dominating mechanism). Let $\mathcal{A}, \mathcal{A}_i : \mathcal{X}^* \to \mathcal{O}, i = 1, \ldots, N$ be randomised algorithms. We call $\mathcal{A}$ a *sum-dominating* mechanism w.r.t. $\mathcal{A}_i, i = 1, \ldots, N$, if

$$\sup_{x \simeq x'} H_\alpha \left( \sum_{i=1}^N \mathcal{A}_i(x_i) \| \sum_{i=1}^N \mathcal{A}_i(x_i') \right) \leq \sup_{x \simeq x'} H_\alpha \left( \mathcal{A}(x) \| \mathcal{A}(x') \right), \tag{8}$$

where $H_\alpha$ is the hockey-stick divergence, and $\simeq$ is the DP neighbourhood relation.

Considering concrete mechanisms that satisfy Definition 4.2, one simple example is given by DP mechanisms that use infinitely divisible noise, as formalised next in Lemma 4.3:

**Lemma 4.3** (Additive mechanisms with infinitely divisible noise are sum-dominated). *Assume $\mathcal{A}_i, i = 1, \ldots, N$ are additive DP mechanisms s.t. they add noise from an infinitely divisible noise family $\Xi$:*

$$\mathcal{A}_i(x_i) = f(x_i) + \xi_i, \tag{9}$$

*where $f$ is some (bounded sensitivity) function, and $\xi_i \in \Xi \, \forall i$. Then the mechanism*

$$\mathcal{A}(x) := \sum_{i=1}^N \left( f(x_i) + \xi_i \right) \tag{10}$$

*is a sum-dominating mechanism w.r.t. $\mathcal{A}_i, i = 1, \ldots, N$.*

*Proof.* Immediately clear by definition of $\mathcal{A}$. □

One prominent example of the possible mechanisms covered by Lemma 4.3 is the ubiquitous continuous Gaussian mechanism:

*Example* 4.4 (Gaussian mechanism). Assume $\mathcal{A}_i$ is a Gaussian mechanism with noise covariance $C^2 \sigma_i^2 \cdot I_d$ and $f$ has bounded sensitivity $C$. Since the normal distribution is infinitely divisible, from Lemma 4.3 it follows that the combined mechanism $\mathcal{A} = \sum_{i=1}^N \mathcal{A}_i$, which is another Gaussian with sensitivity $C$ and noise covariance $C^2 (\sum_{i=1}^N \sigma_i^2) \cdot I_d$, is a sum-dominating mechanism. Finally, due to well-known existing results (see e.g. Meiser & Mohammadi 2018; Koskela et al. 2020; Zhu et al. 2022), a (tightly) dominating pair of distributions $(P, Q)$ in the sense of Definition 3.3 for the sum-dominating mechanism $\mathcal{A}$ is given by a pair of 1d Gaussians with means $\mu_P = 0, \mu_Q = 1$, and variances $\sigma_P^2 = \sigma_Q^2 = \sum_{i=1}^N \sigma_i^2$.

Other mechanisms covered by Lemma 4.3 include existing discrete infinitely divisible noise mechanisms compatible with practical SecAgg protocols, such as Skellam (Valovich & Aldà, 2017; Agarwal et al., 2021), and Poisson-binomial (Chen et al., 2022b).[3]

Next, we consider composing a sum-dominating mechanisms over $S$ (local) steps. This allows us to account for the total privacy when doing more than one local optimization steps:

---

[3]Discrete Gaussian (Canonne et al., 2020) is not infinitely divisible, but is close-enough that a sum-dominating mechanism can still be found in many practical settings, see Kairouz et al. (2021). In such cases, the inequality in Definition 4.2 could always be strict, whereas for any infinitely divisible noise mechanism it can be written as an equality (see Lemma 4.3). We note that even in the infinitely divisible case, however, writing Definition 4.2 with inequality is necessary to avoid nonsensical limitations, such as a having a DP mechanism that satisfies Definition 4.2 with a given $\delta$ while not satisfying it for any $\delta' > \delta$.

**Lemma 4.5.** *Assume $\mathcal{A}^{(s)}$ is a sum-dominating mechanism w.r.t. $\mathcal{A}_i^{(s)}, i = 1, \ldots, N$ for every $s = 1, \ldots, S$. Then the composition of the sum-dominating mechanisms $(\mathcal{A}^{(1)}, \ldots, \mathcal{A}^{(S)})$ dominates the composition*

$$\left( \sum_{i=1}^{N} \mathcal{A}_i^{(1)}, \ldots, \sum_{i=1}^{N} \mathcal{A}_i^{(S)} \right). \tag{11}$$

*Proof.* For any $s \in \{1, \ldots, S\}$, we immediately have

$$\sup_{x \simeq x'} H_\alpha \left( \sum_{i=1}^{N} \mathcal{A}_i^{(s)}(x_i) \| \sum_{i=1}^{N} \mathcal{A}_i^{(s)}(x_i') \right) \leq \sup_{x \simeq x'} H_\alpha \left( \mathcal{A}^{(s)}(x) \| \mathcal{A}^{(s)}(x') \right) \tag{12}$$

by definition of $\mathcal{A}$ (Definition 4.2). The claim therefore follows immediately from (Zhu et al., 2022, Theorem 10). $\qquad\square$

Considering Lemma 4.5, in our case it essentially says that to account for running $S$ local optimization steps, it is enough to find a proper sum-dominating mechanism for each step separately.

With the next result given in Lemma 4.6, we can connect the previous results with the form of output we get from actually running local optimization in FL:

**Lemma 4.6.** *Assume that releasing the vector*

$$\left( \sum_{i=1}^{N} \mathcal{A}_i^{(1)}(x_i), \ldots, \sum_{i=1}^{N} \mathcal{A}_i^{(S)}(x_i) \right) \tag{13}$$

*satisfies $(\varepsilon, \delta)$-DP. Then releasing*

$$\sum_{i=1}^{N} \sum_{s=1}^{S} \mathcal{A}_i^{(s)}(x_i) \tag{14}$$

*also satisfies $(\varepsilon, \delta)$-DP.*

*Proof.* Due to the post-processing immunity of DP (see, e.g., Dwork & Roth 2014), the assumption implies that releasing

$$\sum_{s=1}^{S} \sum_{i=1}^{N} \mathcal{A}_i^{(s)}(x_i) \tag{15}$$

satisfies $(\varepsilon, \delta)$-DP, and by exchanging the order of summation the claim follows. Note that all the mechanisms are assumed to be DP w.r.t. their first argument for any given auxiliary value, which allows us to do the exchange without affecting privacy (in the context of FL, we effectively switch from communicating between each local step to running all local steps and then communicating). $\quad\square$

Taken together, Definition 4.2 or Lemma 4.3 along with Lemmas 4.5 & 4.6 allow us to compose DP mechanisms with joint noise scaling over the clients. In our main result given as Theorem 4.7, we show that each client can run DP-SGD with several local steps while still benefiting from joint noise scaling when communicating the update via a trusted aggregator.

**Theorem 4.7.** *Assume $N$ clients use local noise mechanisms $\mathcal{A}_i^{(s)}, i = 1, \ldots, N$ as in Lemma 4.3 for each local gradient optimization step $s = 1, \ldots, S$, and that the final aggregated update $\sum_{i=1}^{N} \Delta_i$ is released via an ideal trusted aggregator. Then denoting the sum-dominating mechanism for step $s$ by $\mathcal{A}^{(s)}$, the query release satisfies $(\varepsilon(\delta), \delta)$-DP for any $\delta \in [0, 1]$, when $\varepsilon(\delta)$ is given by accounting for releasing the vector*

$$\left( \mathcal{A}^{(1)}(x), \ldots, \mathcal{A}^{(S)}(x) \right),$$

*where $x = \cup_{i=1}^{N} x_i$.*

*Proof.* For privacy accounting, assuming all sums are done by trusted aggregator $TA$, releasing the aggregated update $TA(\sum_{i=1}^N \Delta_i)$ corresponds to releasing the result

$$-\gamma \sum_{i=1}^N \sum_{s=1}^S \mathcal{A}_i^{(s)}(x_i; z_{i,s}),$$

where each mechanism includes a mapping that maps the local samples to the clipped per-unit gradients as well as the DP noise, and $z_{i,s}$ are auxiliary values (e.g., state after the previous step).[4] Since all mechanisms are assumed to be DP w.r.t. the first argument for any auxiliary value, the auxiliary values do not affect the DP guarantees, and hence we do not write them explicitly in the following.

From Lemma 4.6 it follows that valid DP guarantees can be established by accounting for the release of the vector $\left(\sum_{i=1}^N \mathcal{A}_i^{(1)}(x_i), \ldots, \sum_{i=1}^N \mathcal{A}_i^{(S)}(x_i)\right)$. Furthermore, Lemma 4.3 implies that for any step $s \in 1, \ldots, S$, the sum-dominating mechanism $\mathcal{A}^{(s)}$ dominates $\sum_{i=1}^N \mathcal{A}_i^{(s)}$, and therefore by Lemma 4.5 the claim follows. $\square$

We note that while Theorem 4.7 assumes infinitely divisible noise mechanism (as is commonly used with DP-SGD in practice), the result is trivial to generalize to any sum-dominating mechanism $\mathcal{A}$, such as discrete Gaussian (Canonne et al., 2020), by relying on Definition 4.2 instead of using Lemma 4.3.

Considering tightness of the privacy accounting done based on Theorem 4.7, it is worth noting that since the accounting relies on Lemma 4.6, which assumes releasing each local step while the actually released query answer is a sum over the local steps, the resulting privacy bound need not be tight but an upper bound on the privacy budget. However, this matches the usual DP-SGD privacy accounting analysis (see e.g. Mironov et al. 2019; Koskela et al. 2020), which typically needs to account for each local optimization step due to technical reasons even if only the final model is released. In the general case, it has also been shown that hiding the intermediate steps does not bring any privacy benefits compared to the per-step accounting (Annamalai, 2024).

## 5 EXPERIMENTS

**Setup and Motivation:** Our chosen settings try to mimic a typical cross-silo FL setup: there are a limited number of clients, each having a smallish local database. The clients have enough local compute to run optimization on the chosen model, while the number of server-client communications required for updating the global model are the main bottle-neck. Note that this bottleneck will emerge even with larger actual organisations training models with broadband connections, when the model size is large-enough, e.g., when training foundation models (Bommasani et al., 2021). This is especially true when using SecAgg protocols, since the cost of running a real SecAgg algorithm presents significant compute and communication overheads even with the efficient protocols discussed in Appendix A.2. In this setting, it makes sense to try and push more optimization steps to the clients while reducing the number of global updates (FL rounds). We also assume that the clients send their local updates via some trusted aggregator (which we only assume and do not implement in practice in the experiments. However, we do use only discrete DP mechanisms compatible with standard SecAgg algorithms in all the experiments). For more details on all the experiments, see Appendix A.4.

**CNN on Fashion-MNIST:** We first train a small convolutional neural network (CNN) on Fashion MNIST data (Xiao et al., 2017), that is distributed iid among 10 clients. We use the CNN architecture introduced by Papernot et al. (2021); Tramèr & Boneh (2021). Figure 1 shows the mean with standard error of the mean (SEM) over 5 repeats for test accuracy and loss with DP-SGD using Skellam noise (Agarwal et al., 2021) with 32 bits gradient quantization,.i.e., without quantization. We train the model for 20 FL rounds and varying number of local steps. Comparing the results for 1 local step as opposed to 1 local epoch ($\simeq 11$ steps, but with different sampling fraction compared to baseline), it

---

[4]For example, with standard DP-SGD, sample-level DP and continuous Gaussian noise, $\sum_{i=1}^N \Delta_i = -\gamma \sum_{i=1}^N \sum_{s=1}^S (g_{i,s} + \xi_{i,s})$, where $g_{i,s}$ are (sums of) clipped per-sample gradients and $\xi_{i,s}$ are the per-step Gaussian noises.

is evident that being able to take more local optimization steps (as allowed by Theorem 4.7) brings considerable utility benefits under fixed privacy and communication budgets.

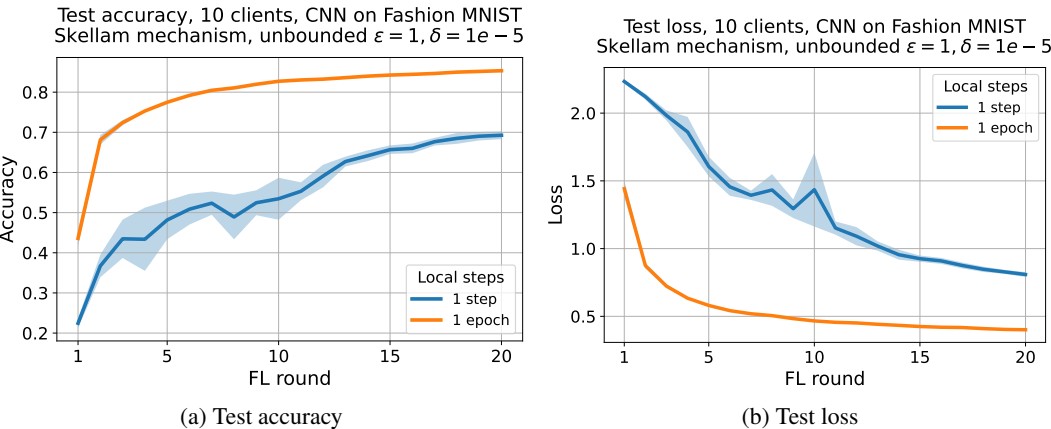

(a) Test accuracy

(b) Test loss

Figure 1: CNN on Fashion-MNIST, 10 clients, mean and SEM over 5 seeds. Running more local steps is clearly beneficial.

**Linear Classifier on Transformed CIFAR-10:** Overall, assuming a fixed privacy budget, we might expect the benefits from being able to run more local steps to be more accentuated with more complex models and very limited communication budget, while for simple-enough models and more FL rounds even a few local steps could lead to good results. To test to what extent this is true for simple yet still useful models, we consider CIFAR-10 data (Krizhevsky, 2009). Similar to Tramèr & Boneh (2021), we take a ResNeXt-29 model (Xie et al., 2017) pre-trained with CIFAR-100 data (Krizhevsky, 2009), remove the final classifier, and use it as a feature extractor to transform the input data. We distribute the transformed CIFAR-10 data iid to 10 clients, and train linear classification layers from scratch for $10, 20, 40, 80$ and $160$ FL rounds using DP-SGD with Skellam noise, 32 bit gradient quantization, and varying number of local steps (1 epoch $\simeq 19$ steps, but with different batch size compared to baseline). The mean and SEM over 5 seeds of the best results for each model over the training run are shown in Figure 2. The benefits of being able to run more than a single local steps are again clear; even with the relatively simple linear model, using 1 local step needs roughly an order of magnitude more FL rounds over a fairly broad range of available communication budgets to reach a similar performance compared to using 1 local epoch.

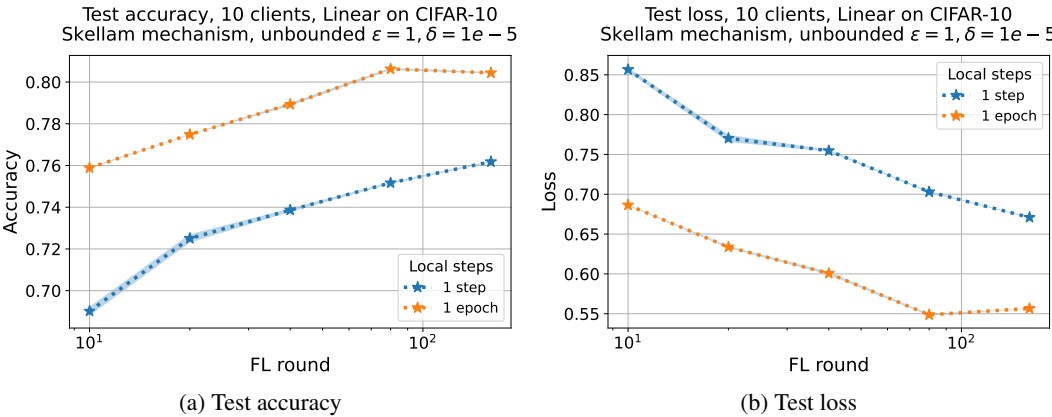

(a) Test accuracy

(b) Test loss

Figure 2: Mean and SEM over 5 seeds of the best performance over training runs for Linear models on CIFAR-10 using pre-trained ResNeXt29 as feature extractor for varying number of FL rounds, 10 clients. Running more local steps is clearly beneficial.

**Logistic Model on Income:** To further test the robustness of the possible benefits from being able to run more than a single local optimization step, we train a simple Logistic Neural Network (LNN) model (i.e., 1-layer fully connected linear classification network similar to the one used in

the previous experiment, but without any pre-trained feature extractor) on ACS Income data (Ding et al., 2021). Unlike the synthetic iid data splits used in the previous experiments, Income data has an inherent client split corresponding to 51 states from where the data has been collected. Since the inherent split is heterogeneous (different states have very number of samples as well as different data distributions), we would expect the benefits of doing more local optimization steps between global communication rounds to dwindle, since the local models from different clients could diverge when only trained locally. However, as shown in Figure 3, even in this setting taking more local steps can be very beneficial (here 1 epoch $\simeq 20$ steps with same local sampling fraction compared to baseline). This clearly demonstrates the utility of our analysis.

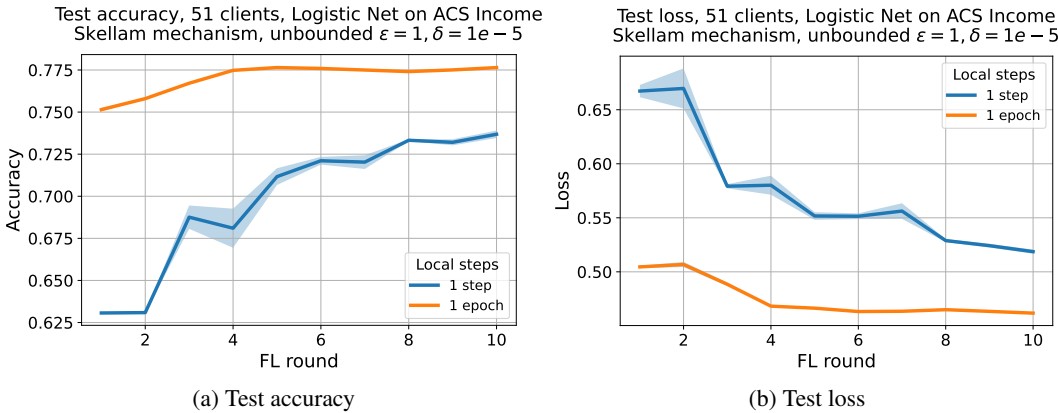

(a) Test accuracy         (b) Test loss

Figure 3: LNN on ACS Income, 51 clients, mean and SEM over 5 seeds. Running more local steps is clearly beneficial.

**Improved Privacy from Model Averaging:** Finally, our results might sometimes be of use also outside standard FL. For example, consider a setting where we have $N$ copies of a model trained on disjoint data sets (e.g., one could think of independent parties learning a classifier on top of a common pre-trained model or fine-tuning a common pre-trained model; either scenario would usually lead to shared model structure and possibly also hyperparameters without explicit coordination), and the parties would like to combine the models post-hoc without running any joint training from scratch. Since this can be seen as FL with a single FL round, if the original model training on each party satisfies Assumption 4.1 (or the relaxed assumptions in Appendix A.1), then a simple averaging of the weights will result in a joint model with improved privacy guarantees against adversaries without access to the original models.

To demonstrate this effect, we account for privacy assuming the same linear model used in Figure 2 (but without actually training any models), Skellam mechanism with 32 bit gradients, Poisson subsampling with sampling probability $0.1$, and varying number of parties and local steps. The accounting is done as it would be done in a realistic setting: we first find a noise level $\sigma_{LDP}$ that results in the target privacy level (unbounded ($\varepsilon = 5, \delta = 1e-5$)-LDP) for each separate model with the chosen number of local steps. We then assume that the local training satisfies Assumption 4.1 and calculate the privacy for averaging varying number of local models. Combining even 2 models results in clearly improved privacy for the averaged model (see Table 1 in Appendix A.5).

## 6 DISCUSSION

In this paper we have shown how to combine multiple local steps in DPFL using fine-grained protection granularities with SecAgg, and empirically demonstrated that this can bring considerable utility benefits under various communication-constrained settings. Our experimental results stand in stark contrasts with the message from the currently existing theoretical bounds for DPFL (Malekmoham-madi et al., 2024, Theorem 3.2), which do not show any benefit from increasing the number of local steps. This disagreement of experimental and theoretical results underlines the need for improved theoretical analysis to understand the conditions under which increasing the number of local steps can lead to improved utility, similar to the recent breakthroughs in analysing non-DP FL (Mishchenko et al., 2022).

**Ethics Statement**    The authors acknowledge that this research, like all research, could have potential negative side-effects. However, given i) that the aim of the paper is strictly to improve techniques available for guaranteeing individual privacy, ii) that we do not introduce any new general purpose algorithms or new data sets, and iii) that we do not see any major ethical problems regarding the objectives or the data sets used in the experiments, we judge the risk of encountering significant negative outcomes from publishing this research to be minimal.

**Reproducibility Statement**    To ensure that the results can be reproduced, the code used in running all the experiments is included as a supplementary material. For the possible camera-ready version of the paper, the full source code will also be published on GitHub with more detailed documentation to try and ensure that replicating all the results and running comparisons should be possible for anyone with relative ease.

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

# A  APPENDIX

## A.1  LOOSENING ASSUMPTIONS

In the main paper, as stated in Assumption 4.1, for each FL round we have assumed constant learning rate $\gamma$, norm clipping bound $C$, noise level $\sigma$, and number of local optimization steps $S$, all of them shared by all the clients. Next, we consider loosening these assumptions. As before, w.l.o.g. we consider only a single FL round, and will therefore omit the index $t$.

Focusing first on the learning rate, we can immediately generalize our results to allow for different learning rate $\gamma_s$ for each local step $s = 1, \ldots, S$: with this notation, following the reasoning of Theorem 4.7, the aggregated update from the clients is given by

$$\sum_{i=1}^{N} \Delta_i = -\sum_{i=1}^{N} \sum_{s=1}^{S} \gamma_s \mathcal{A}_i^{(s)}(x_i; z_{i,s}), \tag{16}$$

which can again be seen as post-processing the vector $\left( \sum_{i=1}^{N} \mathcal{A}_i^{(1)}(x_i), \ldots, \sum_{i=1}^{N} \mathcal{A}_i^{(S)}(x_i) \right)$, so we can again use Lemma 4.6 for accounting without encountering problems.

When considering client-specific learning rates things can be more complicated. The main issue now is to find proper sum-dominating mechanisms that satisfy:

$$\sup_{x \simeq x'} H_\alpha \left( \sum_{i=1}^{N} \gamma_{i,s} \mathcal{A}_i^{(s)}(x_i) \| \sum_{i=1}^{N} \gamma_{i,s} \mathcal{A}_i^{(s)}(x_i') \right)$$
$$\leq \sup_{x \simeq x'} H_\alpha \left( \mathcal{A}^{(s)}(x) \| \mathcal{A}^{(s)}(x') \right), s = 1, \ldots, S. \tag{17}$$

As a concrete example, assume $\mathcal{A}_i^{(s)}$ is the continuous Gaussian mechanism with shared norm clipping constants and noise levels $C_{i,s} = C_s, \sigma_{i,s} = \sigma_s \ \forall i$. Dropping the step index $s$ for readability, let $\gamma_i = \frac{\gamma_1}{l_i}$ for some $l_i > 0, i = 2, \ldots, N$. Writing $g_i$ for a sum over the per-unit clipped gradients of

client $i$, and $\xi_i \sim \mathcal{N}(0, C^2\sigma^2 \cdot I_d)$ a single optimization step now contributes the following term for the global update:

$$-\sum_{i=1}^{N} \gamma_i \left(g_i + \xi_i\right) \tag{18}$$

$$= -\gamma_1 \left(g_1 + \xi_1 + \sum_{i=2}^{N} \frac{g_i + \xi_i}{l_i}\right) \tag{19}$$

$$= -\gamma_1 \left(g_1 + \sum_{i=2}^{N} \frac{g_i}{l_i} + \xi\right), \tag{20}$$

where $\xi \sim \mathcal{N}(0, C^2\sigma^2[1 + \sum_{i=2}^{N} \frac{1}{l_i^2}] \cdot I_d)$, which is a sum-dominating Gaussian mechanism. When accounting for the sum-dominating mechanism, it has sensitivity $C^* = \max\{C, \frac{C}{l_2}, \ldots, \frac{C}{l_N}\}$, which in turn gives noise variance $(\frac{C}{C^*})^2 \sigma^2 [1 + \sum_{i=2}^{N} \frac{1}{l_i^2}]$ for DP.

Similarly, we could relax the assumptions further to allow the clients to use different clipping and noise levels $C_i, \sigma_i$. As before, a single optimization step can again be written in the form of Equation 20, when

$$\xi \sim \mathcal{N}\left(0, \left[C_1^2\sigma_1^2 + \sum_{i=2}^{N} \frac{C_i^2\sigma_i^2}{l_i^2}\right] \cdot I_d\right). \tag{21}$$

For global privacy accounting with a sum-dominating Gaussian mechanism, suitable sensitivity is now given by $C^* = \max\{C_1, \frac{C_2}{l_2}, \ldots, \frac{C_N}{l_N}\}$, and the resulting variance for accounting is $\sum_{i=1}^{N} (\frac{C_i\sigma_i}{C^*})^2$.

Assuming clients have differing number of local steps, we can try to fuse some local steps for the privacy analysis until all clients have the same number of steps $S$, after which we can then use the earlier results.[5]

As a simple example, assume we have 2 clients running DP-SGD: client 1 runs $S$ local steps using norm clipping constant $C$ and Gaussian mechanism with noise variance $\sigma^2$, while client 2 runs $2S$ local steps with clipping $C/2$ and Gaussian noise variance $\sigma^2$. The difference now is that while the clipping is done on each step, from the privacy accounting perspective we can disregard some noise and think that client 2 adds noise only on every other step. Looking at the update from client 2, we would then have

$$\Delta_2 = -\gamma \sum_{s=1}^{2S} (g_{2,s} + \mathbb{I}[s = 2l, l \in \mathbb{N}] \cdot \xi_{2,s}) \tag{22}$$

$$= -\gamma \sum_{s=1}^{S} (g'_{2,s} + \xi'_{2,s}), \tag{23}$$

where $g_{2,s}$ are the clipped per-sample gradients, $g'_{2,s} := g_{2,2s-1} + g_{2,2s}$, $\xi_{2,s}$ are the noise values, $\xi'_{2,s} := \xi_{2,2s}$, and $\mathbb{I}$ is the indicator function. Due to the clipping, the sensitivity of each fused step can be easily upper bounded via triangle-inequality: $\|g_{2,s'}\|_2 = \|g_{2,2s'-1} + g_{2,2s'}\|_2 \leq \|g_{2,2s'-1}\|_2 + \|g_{2,2s'}\|_2 \leq C$. Since Equation 23 now has the same number of local steps as client 1 is taking, we can readily use the previous results to enable privacy accounting for the aggregated update. Combining the fusing of local steps with the previous notes on differing clipping norm values, learning rates and noise variances allows us to use our main results in several settings beyond what is stated in Assumption 4.1.

As a final note, when the clients use data subsampling for the local optimization, differing local subsampling probabilities can lead to having varying DP guarantees between the clients on the global level due to the different subsampling amplification effects, but can otherwise be incorporated with the same analysis we have already presented.

---

[5]Alternatively, we could also consider breaking some local steps into several parts. We leave the detailed consideration of this approach for future work.

## A.2 From Ideal Trusted Aggregators to Practical SecAgg Protocols

For implementing the trusted aggregator assumed in Theorem 4.7 in practice, it should be noted that as the sum over $s$ is done locally by each client during local optimization, it is always trusted as long as the individual clients are, while the sum over $i$ would need to be implemented, e.g., using a suitable SecAgg protocol. Several such algorithms are known, including the ones proposed by Bell et al. (2020); Bonawitz et al. (2017); Sabater et al. (2022); So et al. (2021).

Using a SecAgg protocol will typically also place some extra requirements on the DP mechanisms $\mathcal{A}_i^{(s)}$, since the SecAgg algorithms usually run on elements of finite rings. This precludes continuous noise mechanisms. A viable alternative is to use some suitable discrete noise mechanism, such as Skellam (Agarwal et al., 2021) or Poisson-binomial (Chen et al., 2022b). However, differing from the cases considered in the cited papers, since in our case the clients send model updates instead of single gradients, the finite ring size used in the SecAgg protocol needs to accommodate the model update size: it does no good to use Skellam mechanism with gradient quantization to a small number of bits, if the model weights and the resulting model update $\Delta_i$ for client $i$ still uses 32 bit floats.

## A.3 Privacy Accounting Details

For privacy accounting we utilize Rényi DP (RDP):

**Definition A.1.** (Mironov, 2017) Let $\alpha > 1$ and $\varepsilon > 0$. A randomised algorithm $\mathcal{A} : \mathcal{X}^* \to \mathcal{O}$ is $(\alpha, \varepsilon)$-RDP if for every $x, x' \in \mathcal{X}^* : x \simeq x'$

$$D_\alpha(\mathcal{A}(x) \| \mathcal{A}(x')) \leq \varepsilon,$$

where $D_\alpha$ is the Rényi divergence of order $\alpha$:

$$D_\alpha(P\|Q) = \frac{1}{\alpha - 1} \log \mathbb{E}_{t \sim Q} \left( \frac{p(t)}{q(t)} \right)^\alpha.$$

We do privacy accounting for all the experiments based on RDP. Generally, we account for the privacy of each individual local optimization step with joint noise from all the clients selected for a given FL round. When the clients use Poisson subsampling to sample minibatches (we assume each client uses the same probability for including any individual sample in the minibatch), we use standard RDP privacy amplification results. In practice, we use the RDP accountant implemented in Opacus (Yousefpour et al., 2021), as well as bounds for Skellam mechanism by Agarwal et al. (2021) and tight RDP amplification by Poisson subsampling (Steinke, 2022). We calculate the privacy cost of the entire training run in RDP, and then convert into ADP using (Mironov, 2017, Proposition 3). Note that, as is common in DP research, we do not include the privacy cost of hyperparameter tuning in the reported privacy budgets (see e.g. Tramèr & Boneh 2021 for some reasoning on this practice).

## A.4 Experimental Details

All the experimental settings we use satisfy Assumption 4.1. We use DP-SGD with Skellam mechanism to optimise the local model parameters, and standard federated averaging as the aggregation rule for updating the global model in all experiments. For each centralised data set (combining original train and test sets), we split the data randomly into equal shares, which results in having almost the same data distribution on each client. For hyperparameter optimization with each dataset, we first split each clients' data internally into train and test parts with fractions (.8-.2). For tuning all hyperparameters, we use only the training fraction, and divide it further (.7-.3) into hyperparameter train-validation. We use Bayesian optimization-based approach implemented in Weights and Biases (Biewald, 2020) for hyperparameter tuning, and simulate FL using Flower (Beutel et al., 2020).

In general, when tuning hyperparameters we do 50 hyperparameter tuning runs. For each tuning run, we train the model on hyperparameter training fraction, test on the validation fraction, and try to optimise for the final model weighted validation loss. After finishing the hyperparameter tuning, we re-train the model from scratch 5 times with different random seeds with the best found hyperparameters using the entire original training data and testing on the test fraction. We report the mean and the standard error of the mean (SEM) in all the figures. In Figure 2 we plot the minimum test loss/maximum test accuracy taken over the entire training run.

For the experiments with Fashion-MNIST (Xiao et al., 2017) and CIFAR-10 (Krizhevsky, 2009) data sets, we run hyperparameter tuning separately for each combination of number of local steps {1 step, 1 epoch}, and expected minibatch sizes on the grid {64, 128, 256, 512} using Poisson subsampling.

For Fashion-MNIST the best expected batch sizes found are 512 for 1 local epoch, and 128 for 1 local step.

With CIFAR-10, due to heavy computational cost of hyperparameter tuning, we use a single expected batch size for each configuration of local steps {1 step, 1 epoch} and FL rounds {10, 20, 40, 160}. Concretely, we pick the best expected batch size value from the above grid when using Bayesian optimization to tune all hyperparameters with 20 FL rounds. This results in choosing expected batch size 128 for 1 local step and 256 for 1 local epoch. We then use these values and optimize all other hyperparameters separately for all other FL round settings.

With ACS Income data (Ding et al., 2021), we tune all hyperparameters for each combination of local steps {1 step, 1 epoch} with Poisson subsampling using local sampling probability on the grid {0.4, 0.2, 0.1, 0.05} for 10 FL rounds. We report results on the best found local sampling probabilities (0.05 for both).

For ResNeXt-29 8x64, we used pre-trained weights available from `https://github.com/bearpaw/pytorch-classification`. Our implementation of the Skellam mechanism is based on the implementation from `https://github.com/facebookresearch/dp_compression` Chaudhuri et al. (2022); Guo et al. (2023).

For American Community Survey (ACS) Income data set Ding et al. (2021) we use the data for all the states and Puerto Rico for 2018. The goal is to predict whether an individual has income greater than \$50000. Instead of simulating data splits, we use the inherent splits, i.e., we take each original region (state or Puerto Rico) to be a client.

For training all models, we use a small cluster with NVIDIA Titan Xp, and NVIDIA Titan V GPUs. The total compute time of all the training runs (including debugging) over all GPUs amounts roughly to 30-60 GPU days.

## A.5 ADDITIONAL RESULTS

Table A.5 shows the results from averaging several independently trained LDP models as described in Section 5.

Table 1: Improved privacy for averaged models, Skellam mechanism, 32 bits (no quantization), Poisson sampling with sampling fraction $0.1$, each local model is unbounded $(\varepsilon = 5., \delta = 1e-5)$-LDP. Averaging more models improves on the DP guarantees against adversaries who do not have access to the original models.

| Local steps | Parties | $\sigma_{total}$ | avg model $\varepsilon$ |
|---|---|---|---|
| 1 step | 1 | 0.69 | 5.0 |
| 1 step | 2 | 0.98 | 2.78 |
| 1 step | 5 | 1.54 | 1.22 |
| 1 step | 10 | 2.18 | 0.64 |
| 1 epoch | 1 | 0.90 | 5.0 |
| 1 epoch | 2 | 1.28 | 2.61 |
| 1 epoch | 5 | 2.02 | 1.19 |
| 1 epoch | 10 | 2.85 | 0.72 |
| 5 epochs | 1 | 1.18 | 5.0 |
| 5 epochs | 2 | 1.67 | 2.85 |
| 5 epochs | 5 | 2.64 | 1.55 |
| 5 epochs | 10 | 3.73 | 1.03 |

