# OpenReview forum: "On Joint Noise Scaling in Differentially Private Federated Learning with Multiple Local Steps"
_ICLR.cc/2025/Conference — ICLR 2025 Conference Withdrawn Submission_

### Official Review · Reviewer_sTXg · 2024-11-02

**Soundness:** 3
**Presentation:** 3
**Contribution:** 2
**Rating:** 5
**Confidence:** 4

**Summary:**

This paper provides privacy analysis for differentially private federated learning algorithms involving multi-step local training. The author focuses on the notion of sample-level Joint Differential Privacy using Secure Aggregation. The experimental results show the benefit of training with multiple local steps in federated learning with both IID and non-IID data.

**Strengths:**

1. The author defines a new notion of sum-dominating mechanism for DP methods that are compatible with secure aggregation.
2. The privacy budget is thoroughly discussed and analyzed.

**Weaknesses:**

1. The privacy analysis is quite simple and can be derived without difficulty following Definition 4.2, limiting the significance of the work.
2. The paper aims to analyze the privacy budget, but it is not reflected in the experiment set up. The empirical results only show that training with multiple local steps is better than single step, which is apparent from many previous works. An experimental study on privacy auditing to measure the privacy spent is needed to support the main theoretical argument.

**Questions:**

1. It is unclear to me why the sum-dominating mechanism definition allow for joint DP noise via secure aggregation. To my understanding, the mechanism output needs to be discrete to allow for the secure aggregation protocol.
2. What is the value of the communication budget used in the experiments? If possible, can the author provide a measurement of the total communication cost under a fixed privacy budget?

---

### Official Review · Reviewer_1qZC · 2024-11-02

**Soundness:** 3
**Presentation:** 3
**Contribution:** 1
**Rating:** 3
**Confidence:** 4

**Summary:**

The paper presents a protocol to save communication in differentially private (DP) Federated Learning (FL) combined wit Secure Aggregation (SecAgg). The main idea of the protocol is to reduce communication by performing several local training steps instead of a single one between each SecAgg round, as done by currents works in FL+SecAgg that provide differential privacy.

The current work proves that their approach is differentailly private and empirically show their substantial savings of in terms of communication in a cross silo setting (i.e., in experiments where the number of clients is around 50 at maximum).

**Strengths:**

- The paper is well written and clearly explains key concepts.

- Exploring the full potential of differentially private machine learning in a distributed setting where there is no need of a trusted central party is a strong and realistic model and facilitating its applicability has a potentially high impact in privacy preserving technologies.

**Weaknesses:**

The paper provides major weaknesses, explained below.

# Major Weaknesses

The most important weakness is that the paper seems to lack of substantial contributions:

1- In terms of theory, the core of their improvement relies in Lemma 4.6, stating that if the vector $\left(\sum_{i=1}^N \mathcal{A}^{(1)}_i, \dots ,\sum\_{i=1}^N \mathcal{A}^{(S)}_i\right) $ is $(\epsilon, \delta)$-DP then $\sum\_{i=1}^N \sum\_{s=1}^S \mathcal{A}^{(s)}\_i$ is also  $(\epsilon, \delta)$-DP. This seems trivial from post-processing. I am not really seeing why the privacy analysis in the current setting (of many local steps) cannot use already existing tools.

This also seems to be the case in Lemmas 4.5 and 4.3, which immediately follow from existing theory and in Theorem 4.7 which is a direct instantiation of Lemma 4.6 mentioned above to the FL setting.

2- The current empirical analysis compares single local step vs multiple local step DPFL+SecAgg. The results are what is already expected when increasing the number of local steps in (cross-silo) FL. Therefore, I find that this comparison does not provide new research insights. If increasing the number local steps in the current setting is not substantially challenging for the privacy analysis (as it is argued above and shown in the paper), a very interesting direction would be to show the full privacy potential of FL+SecAgg in a broader comparison of DPFL techniques. This would make the contribution sufficiently substantial and could include, for example, techniques that rely on shuffling[R3] or FTRL related techniques [R1,R2] in aspects such as communication, computation and accuracy.


# Minor
- Page 3: the text between parenthesis "(compare this disagreement of empirical results and theory to the discussion by Mishchenko et al. 2022 on the provable usefulness of local steps in non-DP FL)"  is confusing.
-  Page 10: "different states have very number of samples ..." $\rightarrow$ "different states have very *different* number ... "

# References

[R1] McMahan, H. Brendan, Zheng Xu, and Yanxiang Zhang. "A Hassle-free Algorithm for Private Learning in Practice: Don't Use Tree Aggregation, Use BLTs." arXiv preprint arXiv:2408.08868 (2024).

[R2] Kairouz, Peter, et al. "Practical and private (deep) learning without sampling or shuffling." International Conference on Machine Learning. PMLR, 2021.

[R3] Ghazi, Badih, et al. "Differentially private aggregation in the shuffle model: Almost central accuracy in almost a single message." International Conference on Machine Learning. PMLR, 2021.

**Questions:**

- Please elaborate in weakness 1 described above

---

### Official Review · Reviewer_5M7Y · 2024-11-02

**Soundness:** 3
**Presentation:** 3
**Contribution:** 2
**Rating:** 6
**Confidence:** 3

**Summary:**

The paper analyzes sample-level DP privacy accounting in an FL setting with several local updates. First, the authors show that a naive approach to privacy accounting might lead to big noise contributions at each communication round (Section 3.2). Then, the paper bounds the privacy leak of one communication by the privacy leak of a composition of local steps (Theorem 4.7). To conduct this analysis, the authors introduce the concept of sum-dominating mechanisms (Definition 4.2) and study their properties (Lemmas 4.3, 4.5, 4.6). Finally, Section 5 validates the advantage of several local steps for various ML tasks.

**Strengths:**

- The research question is relevant since communication is a frequent limitation of FL protocols.
- The research question is well-motivated by Section 3.2.
- The analysis of sample-level DP with several local steps seems novel.
- The analysis and proofs are easy to understand.
- The paper is well-written and clear.

**Weaknesses:**

- The practical significance of sample-level DP guarantees is hard to judge. Since DP guarantees are quite abstract, it is hard to understand how client-level guarantees differ from sample-level guarantees.
- Theorem 4.7 lacks a comparison with the motivating Section 3.2.
- While Section 5 shows the advantages of several local steps, it does not directly demonstrate the benefit of Theorem 4.7. Specifically, the current experiments conflate two variables: the number of seen examples and the privacy accounting.

**Questions:**

- How do sample-level guarantees differ from client-level guarantees regarding membership inference attacks and reconstruction attacks?
- How does Theorem 4.7 compare with the naive accounting from Section 3.2?
- Can the authors repeat some experiments from Section 5 with more naive privacy accounting (e.g., from Section 3.2)?
- Can the authors explain why Theorem 3.2 of Malekmohammadi et al. (2024) does not show the benefits of more local steps? As I understand it, the privacy accounting in their case results in a quite complicated term $\Psi_\sigma$, which I can not interpret easily.

---

### Note · Authors · 2024-11-13

I have read and agree with the venue's withdrawal policy on behalf of myself and my co-authors.